# Oyster Shell Modified Tobacco Straw Biochar: Efficient Phosphate Adsorption at Wide Range of pH Values

**DOI:** 10.3390/ijerph19127227

**Published:** 2022-06-13

**Authors:** Menghan Feng, Mengmeng Li, Lisheng Zhang, Yuan Luo, Di Zhao, Mingyao Yuan, Keqiang Zhang, Feng Wang

**Affiliations:** 1Agro-Environmental Protection Institute, Ministry of Agriculture and Rural Affairs, Tianjin 300191, China; 82101215307@caas.cn (M.F.); lmm201217@163.com (M.L.); 82101202148@caas.cn (L.Z.); ly234132@163.com (Y.L.); 82101192129@caas.cn (D.Z.); y18235780314@163.com (M.Y.); zhangkeqiang@aepi.org.cn (K.Z.); 2Erhai Watershed Ecological Environment Quality Testing Engineering Research Center of Yunnan Provincial Universities, West Yunnan University of Applied Sciences, Dali 671004, China; 3Dali Comprehensive Experimental Station of Environmental Protection Research and Monitoring Institute, Ministry of Agriculture and Rural Affairs, Dali 671004, China; 4College of Resources and Environment, Yunnan Agricultural University, Kunming 650201, China

**Keywords:** waste management, eutrophication, phosphorous adsorption, livestock wastewater

## Abstract

In order to improve the phosphate adsorption capacity of Ca-loaded biochar at a wide range of pH values, Ca (oyster shell) was loaded as Ca(OH)_2_ on the tobacco stalk biochar (Ca-BC), which was prepared by high-temperature calcination, ultrasonic treatment, and stirring impregnation method. The phosphorus removal performance of Ca-BC adsorption was studied by batch adsorption experiments, and the mechanism of Ca-BC adsorption and phosphorus removal was investigated by SEM-EDS, FTIR, and XRD. The results showed that after high-temperature calcination, oyster shells became CaO, then converted into Ca(OH)_2_ in the process of stirring impregnation and had activated the pore expansion effect of biochar. According to the Langmuir model, the adsorption capacity of Ca-BC for phosphate was 88.64 mg P/g, and the adsorption process followed pseudo-second-order kinetics. The Ca(OH)_2_ on the surface of biochar under the initial pH acidic condition preferentially neutralizes with H^+^ acid-base in solution, so that Ca-BC chemically precipitates with phosphate under alkaline conditions, which increases the adsorption capacity by 3–15 times compared with other Ca-loaded biochar. Ca-BC phosphate removal rate of livestock wastewater (pig and cattle farms) is 91~95%, whereas pond and domestic wastewater can be quantitatively removed. This study provides an experimental basis for efficient phosphorus removal by Ca-modified biochar and suggesting possible applications in real wastewater.

## 1. Introduction

Phosphorus (P) is one of the three essential nutrients (N, P, and K), which plays a pivotal role in maintaining crop growth and improving grain yield [1,2]. The sources of phosphorus emission mainly include three aspects: industrial, agricultural, and domestic sources. However, on the other hand, according to The Second National Pollution Source Census Bulletin issued in 2020, phosphorous ranked 4th among the pollutants discharged in water, with China’s total discharge of phosphorus being 315,400 tons [3]. Excessive inflow of phosphorus into water bodies can result in eutrophication, thus leading to water hypoxia, death of aquatic plants and animals, and damage to water quality. All these result in economic losses and even threaten human health in the affected areas [4,5].

Among the multitude of water phase phosphate separation methods, the adsorption method is a promising method because of its simple operation, environmental friendliness, and high removal rate [6,7,8]. Thus, the development of cost-effective adsorbent materials having excellent adsorption performance has become an important topic of research [9,10]. Phosphate ion, which is usually negatively charged in solution, can act as a Lewis acid and can form coordination complexes with most metals (Lewis bases). So a host of metallic adsorbents have been developed for the aqueous phase removal of phosphorus resources [11,12]. Among them, Ca, which is a cheap, environmental friendly, and widely abundant metal in the earth’s crust, has been found to be efficient for phosphate removal from solution [13,14,15,16]. Therefore, Ca loaded on other carrier materials can be effective sorbent materials for the recovery of phosphorus resources from wastewater.

Biochar is a carbon-rich material, which can be obtained by pyrolysis of biomass under anoxic conditions. It is widely available, with simple preparation methods and large specific surface area, and is also suitable as a good carrier [17,18,19,20]. Ca^2+^ is the form in which Ca is often loaded onto biochar; however, the pH of the water has great influence on the adsorption of phosphate by Ca^2+^-loaded biochar, and the adsorption capacity at lower pH range is usually poor [21,22]. Feng et al. obtained a maximum adsorption capacity of only 6 mg P/g for Ca^2+^-loaded composites of sheep manure biochar when the solution pH was less than 7 [12]. Ca^2+^-loaded hyacinth (*Eichhornia crassipes*) biochar and sludge biochar prepared by Ramirez-Muñoz and Antunes showed respective maximum adsorption capacities of 17 mg P/g, and 36 mg P/g were obtained for a solution having a pH < 7 [23,24]. Due to ineffective binding of Ca^2+^ and PO_4_^3−^ in the low pH region, the adsorption capacity of Ca^2+^ loaded biochar is poor and thus limiting its application range. Additionally, in recent years, Ca(OH)_2_-loaded nanoparticles as an antacid agent can not only enhance the phosphate adsorption capacity under low pH conditions [25,26], but also can be widely used in the protection of artificially aged wood lignin under acidic atmosphere as well as cultural heritage [27,28].

With the above background, Ca was loaded as Ca(OH)_2_ onto the biochar obtained from tobacco straw and oyster shell in order to improve the adsorption capacity of Ca-loaded biochar towards phosphate under acidic condition, so as to achieve the phosphorus adsorption of the loaded Ca biochar at wide pH. The biochar was prepared by high temperature calcination-ultrasonic treatment-stirring impregnation method using tobacco straw and oyster shell as the raw materials. Batch adsorption experiments were carried out to investigate the phosphorus removal effect of Ca-BC in simulated wastewater. The morphology of the synthesized Ca-BC was investigated by various characterization methods, which were also used to investigate the adsorption mechanism. The phosphorus removal effect of Ca-BC in actual wastewater was investigated by adsorption experiments from the wastewater of cattle farm and pig farm breeding water as well as pond and domestic sewage.

## 2. Materials and Methods

### 2.1. Materials

Tobacco straws taken from a tobacco planting plot in Xizhou Town, Dali City, Yunnan Province were cleaned, crushed, sieved, and bagged with 60–100 mesh sieves; oyster shells produced in Beihai City, Guangxi Zhuang Autonomous Region, were cleaned, grounded, sieved, and bagged with 100 mesh sieves. Potassium dihydrogen phosphate was purchased from Tianjin Windship Chemical Reagent Technology Co., Ltd., Tianjin, China. The potassium dihydrogen phosphate used in this study was of very high purity (purity level: KH_2_PO_4_ ≥ 99.5%), and all other chemical reagents were of analytical purity. Deionized water with a resistivity of 18.2 MΩ·cm^−1^ was used in all the experiments.

### 2.2. Biochar Preparation and Modification

A total of 70 g of sieved tobacco straw was transferred to a crucible, which was covered with a lid to isolate the air and then was placed in a muffle furnace for raising the temperature to 500 °C at a rate of 5 °C/min, after which it was held for 2 h. On the other hand, 20 g of sieved oyster shell was transferred to a crucible, the temperature of which was raised to 1000 °C at a rate of 5 °C/min in a tube furnace and then held for 1 h. Thereafter, 10 g of calcined oyster shell was dissolved in 500 mL of water and sonicated for 1.5 h to obtain a suspension of Ca(OH)_2_. Thereafter, 20 g of pyrolyzed biochar was added to the solution and stirred at 800 rpm/min for 6 h at room temperature in a constant temperature magnetic stirrer. Then, the mixture was transferred to an oven at 105 °C for drying and removal of water, thereby obtaining a Ca(OH)_2_-loaded biochar composite, which was named as Ca-BC [29,30]. The pyrolyzed straw biochar only was named BC.

### 2.3. Adsorption Experiments

#### 2.3.1. Adsorption Isotherms

A constant amount of Ca-BC (0.05 g) was weighed into a series of 100 mL conical flasks in which 50 mL of KH_2_PO_4_ solution having different concentrations (5, 10, 15, 25, 50, 75, 100, 150, 200 mg P/L) was added individually. Then, the mixtures were shaken in a constant temperature shaker at 180 rpm for 24 h at room temperature (25 ± 0.5) °C. After the equilibrium was reached, supernatant solution was separated via filtration by passing through 0.45 μm microporous membrane. The phosphorus concentration in the supernatant solution after filtration was determined using spectrophotometry employing ammonium molybdate as the chromogenic reagent, the wavelength used to measure P concentration is 700 nm. Each experiment was repeated 3 times, and the mean value of all the experiments was reported as the obtained value. The test data were fitted using Langmuir and Freundlich adsorption isotherms [29,30]. The isothermal adsorption equations are shown below:(1)$$\text{Langmuir adsorption isothermal equation: }q_{e}=q_{max}\frac{K_{L}C_{e}}{1+K_{L}C_{e}}$$
(2)$$\text{Freundlich adsorption isothermal equation: }q_{e}=K_{F}C_{e}{}^{\frac{1}{n}}$$

Here, *K_L_* (L/mg) is the adsorption equilibrium constant for Langmuir adsorption isotherm; *K_F_* (mg^(1−1/*n*)^ L^1/*n*^/g) and n are Freundlich’s adsorption capacity and adsorption strength constants, respectively; *q_e_* denotes the equilibrium adsorption capacity (mg/g); *q_max_* denotes the maximum adsorption capacity (mg/g); *C_e_* is the equilibrium concentration of phosphate in solution (mg/L).

#### 2.3.2. Adsorption Kinetics

0.05 g Ca-BC was weighed into 100 mL separate conical flasks and in each flask 50 mL of KH_2_PO_4_ solution having 150 mg P/L was added, followed by shaking for predetermined time intervals (10 min, 30 min, 60 min, 120 min, 180 min, 240 min, 480 min, 720 min, and 1440 min). Other parameters were same as mentioned above. The test data were fitted using pseudo-first-order and pseudo-second-order kinetics [24,31]. The kinetic equations are shown below.
(3)$$\text{Pseudo-first-order kinetic equation: }q_{t}=q_{e}\left(1-e^{-k_{1}t}\right)$$
(4)$$\text{Pseudo-second-order kinetic equation: }\frac{t}{q_{t}}=\frac{1}{k_{2}q_{e}{}^{2}}+\frac{1}{q_{e}}t$$

Here, *q_t_* (mg/g) is the amount of phosphorus adsorbed at time *t*; *q_e_* (mg/g) is the amount of phosphorus adsorbed at equilibrium; *k*_1_ (min^−1^) is the adsorption rate constant for quasi pseudo-first-order equation; *k*_2_ (g/(mg·min)) is the adsorption rate constant for pseudo-second-order equation.

#### 2.3.3. Effect of pH and Coexisting Anions on the Phosphate Adsorption

A total of 0.05 g of Ca-BC was weighed in different conical flasks with each having volume of 100 mL, and in each flask 50 mL of KH_2_PO_4_ solution having different pH values of 3, 4, 5, 6, 7, 8, 9, 10, and 11 (150 mg P/L) was added. A total of 1 mol/L HCl or NaOH was used to adjust the pH values of KH_2_PO_4_ solution, and then each conical flask was shaken for 24 h. All the other parameters were as mentioned above.

To understand the effect of co-existing anions, 0.05 g of Ca-BC was weighed in a 100 mL conical flask into which 50 mL of KH_2_PO_4_ solution consisting of Cl^−^, NO_3_^−^, CO_3_^2−^ and SO_4_^2−^ with each having concentration of 150 mg/L was added. KCl, KNO_3_, K_2_CO_3_, and K_2_SO_4_ were used as the source of Cl^−^, NO_3_^−^, CO_3_^2−^, and SO_4_^2−^, respectively. Then, the flask was shaken at room temperature for 24 h at 180 rpm. All other details were similar to those described above.

### 2.4. Actual Wastewater Adsorption Experiments

The breeding wastewater from cattle farm and pig farm, pond effluent, and domestic wastewater were selected for the actual wastewater adsorption experiments. The cattle farm wastewater, pond effluent, and domestic sewage were taken from an experimental station in Dali City, Yunnan Province, having pH 7.53 ± 0.1, 8.45 ± 0.1 and 8.13 ± 0.1, respectively, and respective total phosphorus concentrations of 115 ± 2.5, 0.325 ± 0.025, and 0.15 ± 0.025 mg/L. The pH of the pig farm wastewater taken from a farm in Dali, Yunnan Province was 7.15 ± 0.1 and total phosphorus concentration was 78.75 ± 3.75 mg/L. Two sets of 100 mL (five in each set) conical flasks were taken, and in each set 0.1, 0.2, 0.5, 1.0, and 2.0 g of Ca-BC were added individually followed by the addition of 50 mL cattle farm wastewater in one set of conical flasks and 50 mL pig farm wastewater in another set of conical flasks. In another two sets of conical flasks (five conical flasks in each set), 0.01, 0.02, 0.05, 0.07, and 0.1 g of Ca-BC were weighed individually, followed by the addition of 50 mL pond wastewater in each flask of one set and 50 mL of domestic wastewater in each flask of another set. All the conical flaks were shaken for 24 h with all other parameters being identical to those described above.

### 2.5. Characterization and Analytical Method

The specific surface area, pore volume, and mean pore size were obtained by a fully automated physisorption instrument (BET, Micromeritics ASAP 2020, Norcross, GA, USA). The surface morphology of adsorbent material was observed by field emission scanning electron microscopy (SEM, Zeiss Sigma 500, Oberkochen, Germany), and surface elemental distribution of the adsorbent was observed by energy dispersive spectrometry (EDS, Zeiss Sigma 500, Oberkochen, Germany). The zeta potential was observed by Dynamic Light Scattering (DLS, Malvern zs90, England, UK). Fourier transformed infrared spectroscopy (FTIR, Thermo Fisher Nicolet Is5, Waltham, MA, USA) and X-ray diffractometer (XRD, Bruker D8 Advance X-ray diffractometer, Karlsruhe, Germany) were used to observe the changes of functional groups on the adsorbent surface and the diffraction patterns, respectively.

## 3. Results and Discussion

### 3.1. Basic Physicochemical Properties of Biochar

The adsorption performance of biochar largely depends on its specific surface area and porosity. The larger the specific surface area and higher the porosity, the more surface adsorption sites there will be and the higher the adsorption performance will be [32]. BC showed specific surface area of 3.5251 m^2^/g, pore volume of 0.006084 cm^3^/g, and an average pore size of 10.6364 nm; the corresponding values for Ca-BC were 6.5792 m^2^/g, 0.033904 cm^3^/g, and 25.0623 nm, respectively. In comparison to BC, the specific surface area, pore volume, and average pore size of Ca-BC increased by 86.7%, 457.3%, and 135.6%, respectively. The significant increase of different surface properties of Ca-BC is due to the conversion of CaO generated after high-temperature calcination of oyster shell into Ca(OH)_2_ in the aqueous phase, which is alkaline and has an activating and pore-expanding effect on the biochar [33,34,35].

### 3.2. Adsorption of Phosphate onto the Prepared Ca-BC Samples

#### Adsorption Isotherms and Adsorption Kinetics of Phosphate

Results from the fitting of different isotherm models on the adsorption of phosphate by Ca-BC are shown in Figure 1a. It is evident from the figure that with increasing concentration, the rate of adsorption initially increased rapidly, and then decreased slightly to reach equilibrium. Langmuir adsorption isotherm (R^2^ = 0.98) showed better fitting of the isotherm data compared to that of Freundlich (R^2^ = 0.78), thus indicating unimolecular layer adsorption of phosphate on the Ca-BC. The maximum adsorption capacity obtained by Langmuir fitting was 88.64 mg P/g and was found to very close to the experimentally obtained adsorption capacity of 87 mg P/g. In the Freundlich equation, 1/*n* is considered to be an index of the degree of difficulty for adsorption. The adsorption is easier when 1/*n* is between 0.1 and 0.50, but if the value is more than 2, the adsorption is difficult. The 1/*n* value for the Ca-BC is 0.25, as can be seen from Table 1, indicating that it is easier for the as-obtained Ca-BC to adsorb phosphate. The kinetic plot for adsorption of phosphate by Ca-BC is shown in Figure 1b, with the fastest adsorption rate being observed in the first 2 h, which slowed down subsequently from 2–5 h to finally reach equilibrium after 5 h. The pseudo-second-order kinetic equation (R^2^ = 0.99) fits the experimentally obtained results better than the pseudo-first-order (R^2^ = 0.92), indicating that chemisorption is the main controlling parameter for the adsorption of phosphate by Ca-BC. The fast adsorption rate in the initial period till 2 h was due to the presence of a large number of active adsorption sites on the adsorbent surface, and within 2–5 h the adsorption sites were gradually decreased. After 5 h, the adsorption sites tended to saturate when there was repulsion between the adsorbed phosphate and free phosphate in solution and the adsorption reached saturation.

### 3.3. Effects of pH and Coexisting Anions on Phosphate Adsorption

#### 3.3.1. Effect of Initial Solution pH

The effect of initial solution pH on the adsorption capacity of Ca-BC is shown in Figure 2a. As can be seen from the figure, within the pH range of 3 to 11, Ca-BC showed high adsorption capacity (in the range of 75~96 mg P/L) for phosphate. Under acidic conditions, the adsorption capacity varied within 93~96 mg P/L, which decreased to the range of 75–88.5 mg P/L under alkaline conditions. Generally, Ca^2+^-loaded biochar showed poor adsorption capacity for phosphate under acidic conditions with the adsorption capacities of common Ca^2+^-loaded biochar in phosphate solutions having pH < 7 being only 6 mg P/L ~36 mg P/L [12,23,24]. Thus, Ca-BC showed 3~15 times higher adsorption capacity for phosphate under acidic conditions in comparison to Ca^2+^-loaded biochar. The reason for significantly stronger adsorption capacity of Ca-BC under acidic conditions than Ca^2+^-loaded biochar is the presence of a large amount of Ca(OH)_2_ on the surface of Ca-BC material. After the addition of the phosphate solution, the pH of the neutralized solution increases to 8~9 due to acid-base neutralization reaction between the OH^−^ of Ca(OH)_2_ and H^+^ of the solution [18,36]. Therefore, in a phosphate solution having pH < 7, chemical precipitation of Ca-BC takes place in a more alkaline environment unlike the Ca^2+^-loaded biochar.

In addition, according to the surface complexation model, the relationship between the isoelectric point and specific surface area of Ca-BC and its surface potential and surface charge density can be established to obtain the surface charged properties of Ca-BC at different pH. Hydroxyl surface charge density due to the migration of Ca-BC surface protons in solution, the surface charge density changes as the material adsorbs ions from the solution. The pH_PZC_ value of Ca-BC is 3.81 (Figure 2b), which indicates electrostatic inhibition effect on phosphate solutions with initial pH varying from 3.81 to 11 [1,30]. It is speculated that when the initial pH is acidic, chemical binding energy plays a dominant role and phosphate adsorption capacity is high. As the initial pH increased, the electrostatic inhibition increased, and electrostatic repulsion became the main force leading to relatively low adsorption capacity of phosphate. Due to the change in the main driving force between acidic and alkaline conditions, the phosphate adsorption capacity under acidic conditions is slightly higher than that under the alkaline conditions.

#### 3.3.2. Effect of Coexisting Anions

The effect of several common coexisting ions on the phosphate adsorption by Ca-BC is shown in Figure 3. The adsorption capacity of Ca-BC for phosphate adsorption was almost unaffected by the presence of Cl^−^, NO_3_^−^, and SO_4_^2−^. Compared with blank control, the presence of CO_3_^2−^ had a significant effect on the adsorption capacity resulting in a decrease of 15.8% from 87.5 mg P/L to 74.5 mg P/L. It is assumed that CO_3_^2−^ combines with Ca^2+^of Ca(OH)_2_ to form CaCO_3_ precipitate, which becomes fixed on the surface and thus blocks the phosphate nucleation site [37,38]. High adsorption capacity of Ca-BC for phosphate even in the presence of several common ions indicates its significant potential in practical wastewater application.

### 3.4. Batch Adsorption of Actual Wastewater Solutions

Figure 4 presents the results of phosphate adsorption by adding different masses of Ca-BC to 50 mL of four different actual wastewater solutions. The removal rate increased significantly with increasing amount of the material. The treatment results for cattle farm wastewater (Figure 4a) showed that for addition of 2 g Ca-BC, the phosphorus content was reduced from 115 mg P/L to 10 mg P/L with 91% removal rate. Similarly, the phosphorus content in the pig farm wastewater was reduced from 78 mg P/L to 5 mg P/L with 95% removal rate. For the cattle farm wastewater, the phosphorous removal rate increased from 73% to 90% when the dosage was increased from 0.1 g to 1 g, with further increase in the dosage from 1 g to 2 g having no significant effect. This indicates that the addition of 1 g of Ca-BC to 50 mL of solution can have maximum removal rate, with further increase of dosage being wastage. Therefore, from an economical point of view, for 115 mg P/L concentration of phosphorous the reference value for Ca-BC addition is 20 g/L. For pig farm wastewater, the removal rate can reach 91% for the addition of 0.5 g Ca-BC. Therefore, for a concentration of 78 mg P/L, the reference value of Ca-BC addition is 10 g/L. The results of pond wastewater and domestic wastewater treatment show that (Figure 4b) the phosphorus content in both the wastewater solution reached to below detection limit after the addition of 0.1 g Ca-BC with the removal rate being ~100%. No significant decrease in the removal rate of phosphorous was observed when the dosing was 0.07 g and thus in practical application, for 0.15~0.317 mg P/L as the concentration standard, the reference value of Ca-BC addition is 3.5 g/L.

### 3.5. The Adsorption Mechanisms of Ca-BC Material for Phosphate

#### 3.5.1. SEM-EDS and ICP-OES Analysis

Before loading of Ca, the biochar BC showed a rod-like structure as a whole, and magnified image of SEM showed the presence of a smooth biochar surface along with some wrinkles, which might be associated with the morphology of the biomass itself (Figure 5a,b). After Ca loading, the biochar Ca-BC showed an overall rectangular square lamellar structure, and the magnified image showed the rough surface of the biochar with increased pores, and showed a granular or mycelium-like structure on the surface of biochar (Figure 5c,d) due to large amount of Ca(OH)_2_ (according to FTIR and XRD analysis in Section 3.5.2 and Section 3.5.3), which was loaded on the surface of biochar. After phosphate adsorption, the overall structure of Ca-BC-P was a rectangular square but with more obvious lamellae compared with Ca-BC. The magnified image of the Ca-BC-P surface showed agglomerated behavior of the material with the pores and surface being covered by flocculent deposits (Figure 5e,f). The EDS analysis and elemental distribution results are shown in Figure 5 and Table 2. Before the loading of Ca, C and O were found to be the major chemical components on the surface of BC along with a small amount of Ca, which might be related to the biomass itself; after the loading of Ca, C, O, and Ca were found to be the major elements on the surface of Ca-BC, and the loading of Ca based on EDS analysis was found to be smaller than that obtained from the ICP-OES analysis (Table 3), indicating the loading of Ca both on the surface and pores of biochar. After phosphorous adsorption, C, O, Ca, and P were found to be the main elements present on the surface of Ca-BC-P, with the loading of P being less than that obtained from the ICP-OES analysis. This indicated bonding of P took place with Ca in both the surface and pore structure.

SEM-EDS analysis showed significant changes in the overall morphology of Ca-BC compared to that of BC, along with a significant increase in volume, which was probably due to the deposition of a large number of Ca(OH)_2_ particles on the surface of biochar. Moreover, it was also found that both the number of pores on the surface and pore size of Ca-BC increased, which were probably caused by the loading of Ca(OH)_2_ deposited on the surface of biochar [28,31]. Ca(OH)_2_ deposited on the surface of biochar may have played a role in activating the pore expansion during the loading process. After phosphate adsorption, there were a large number of nanoscale flocs, and the EDS mapping showed that elemental P was bound to the biochar. Comprehensive thermodynamic and kinetic analyses showed that Ca-BC and P generated new substances through a complex chemical reaction [39,40].

#### 3.5.2. FTIR Analysis

To investigate the effect of Ca loading and phosphate adsorption on the functional groups of biochar, FTIR analysis was performed on BC, Ca-BC, and Ca-BC-P as shown in Figure 6. The absorption peaks at 1420–1442 cm^−1^, which can be attributed to the C=O stretching vibration for BC, Ca-BC, and Ca-BC-P may be originated from the organic functional groups present in the biochar itself [15,41]. Absorption peaks at 3423 cm^−1^ and 1581 cm^−1^ can be attributed to the -OH stretching and bending vibrations, respectively [8,42]. The characteristic peak corresponding to out-of-plane bending of C-O appears at 873 cm^−1^ [43]. There were significant changes in the absorption peaks of Ca-BC in comparison to that of BC, with a distinct strong and narrow -OH stretching vibration band appearing at 3641 cm^−1^, presumably originating from Ca(OH)_2_, which was successfully loaded onto BC [44]. Comparative analysis of the FTIR spectra before and after phosphate adsorption shows that the -OH absorption peak at 3641 cm^−1^ for Ca-BC-P disappears, while characteristic peak appeared at 1034 cm^−1^ corresponding to the telescopic vibration of HPO_4_^2−^ along with the characteristic peaks at 959 cm^−1^, 602 cm^−1^, and 564 cm^−1^ corresponding to the telescopic vibrations of PO_4_^3^^−^. These peaks corresponding to phosphate ions combined with XRD showed that the biochar Ca-BC and phosphate reacted to form Ca_3_(PO_4_)_2_ precipitate [45]. The absorption peak at 1130 cm^−1^ is attributed to C-N stretching vibrations in addition to C-H and N-H bending vibrations, which may be related to the organic functional groups of biochar [7,46].

#### 3.5.3. XRD Analysis

To investigate the effect of Ca loading and phosphate adsorption on the phase compositions of biochar, XRD analysis was performed on BC, Ca-BC, and Ca-BC-P. The data were analyzed and processed by jade 6.0, and the results are shown in Figure 7. The main characteristic peak of BC was due to the CaCO_3_ phase (PDF#99-0022). The main characteristic peaks of Ca-BC were from the Ca(OH)_2_ (PDF#04-0733) and CaCO_3_ (PDF#99-0022) phases. Thus, phase compositions of the biochar before and after the loading of Ca indicated successful loading of Ca(OH)_2_ on the biochar. For Ca-BC-P, the major peak was corresponding to the phase Ca_3_(PO_4_)_2_, indicating the formation of Ca_3_(PO_4_)_2_ precipitate due to the chemical reaction between phosphate and Ca, which was responsible for the removal of phosphorus from the water, in agreement with the kinetic, SEM, and FTIR findings.

## 4. Conclusions

In this study, Ca(OH)_2_-loaded tobacco straw biochar Ca-BC was prepared using tobacco straw and oyster shell as raw materials, and the phosphate adsorption characteristics of Ca-BC were investigated by conducting batch adsorption experiments to verify the phosphorus removal effect of Ca-BC in actual wastewater, and the main conclusions were as follows:

Langmuir adsorption isotherms (R^2^ = 0.98) fitted the observed results better than the Freundlich isotherm (R^2^ = 0.78), with the maximum phosphorus adsorption capacity reaching 88.64 mg P/g. The equilibrium for adsorption was attained at about 5 h and the pseudo-second-order kinetic model (R^2^ = 0.99) is found to fit the results better than the pseudo-first-order kinetic model (R^2^ = 0.92). The adsorption behavior of Ca(OH)_2_ was probably due to chemisorption via the formation of a monolayer. Ca-BC removed aqueous phosphate by chemical precipitation and electrostatic interaction with phosphate to form Ca_3_(PO_4_)_2_.

Ca-BC maintains good adsorption capacity for phosphate over a wide pH range. Phosphorus adsorption capacities ranging from 75 mg P/L to 96 mg P/L within the pH range of 3 to 11. Under acidic initial pH conditions (pH = 3–6), Ca-BC adsorption capacity reaches 93–96 mg P/g, which is 3–15 times higher than that of Ca^2+^-loaded biochar, mainly because Ca(OH)_2_ on the Ca-BC surface reacted preferentially with H^+^ in solution, allowing Ca-BC to adsorb phosphate in an alkaline environment; under the initial alkaline pH conditions (pH = 7–11), the adsorption capacity was 75–88.5 mg P/g, which was the same as that of Ca^2+^-loaded biochar.

Ca-BC was found to be very effective for the treatment of actual wastewater. Taking pig and cattle farms as an example, the removal rate of phosphate in actual farm wastewater is 91–95% with near quantitative removal of phosphate from the pond and domestic wastewater.

This study provides an experimental basis for efficient phosphorus removal from Ca-modified biochar under acidic conditions; and it offers the possibility of practical production of Ca-modified biochar since no acid is used in the preparation.

## Figures and Tables

**Figure 1 ijerph-19-07227-f001:**
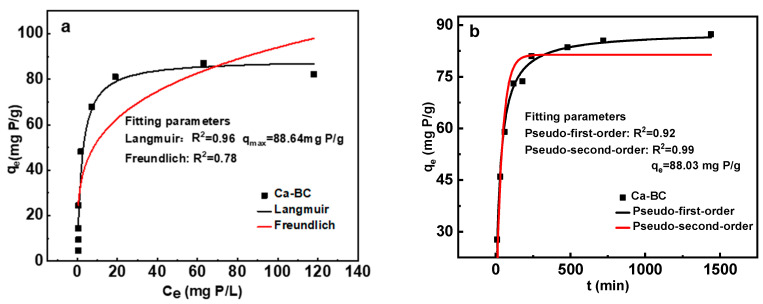
The adsorption isotherm of Ca-BC (**a**). (Dosage: 0.05 g; Temperature: 25 °C; Time: 24 h; pH = 7); The adsorption kinetics of phosphate onto Ca-BC (**b**). (Dosage: 0.05 g; Temperature: 25 °C; Initial phosphorus.

**Figure 2 ijerph-19-07227-f002:**
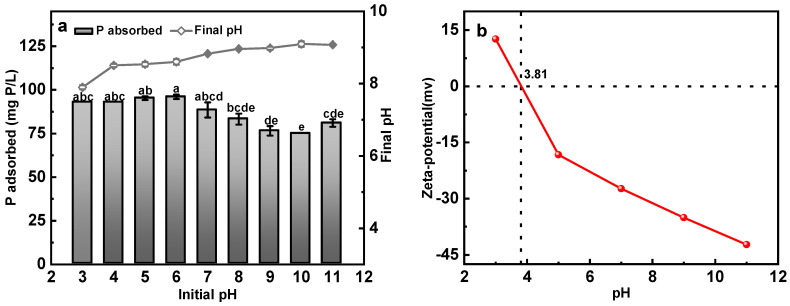
The effect of initial pH on phosphate adsorption on Ca-BC (**a**) (Dosage: 0.05 g; Temperature: 25 °C; Time: 24 h; Initial phosphorus concentration: 150 mg P/L). Zeta potentials of Ca-BC (**b**). The letters a, b, c, d, e represent the results of significance analysis of variance.

**Figure 3 ijerph-19-07227-f003:**
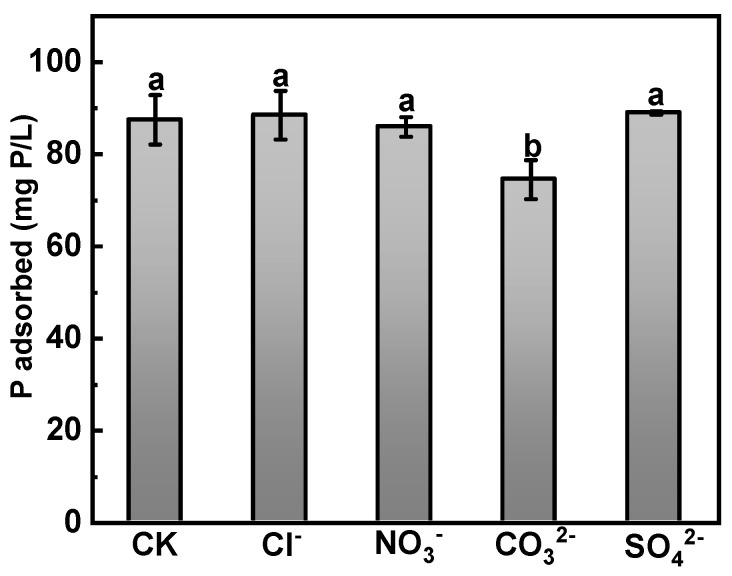
Effect of coexisting anions on phosphate sorption (Dosage: 0.05 g; Temperature: 25 °C; Time: 24 h; Initial phosphorus concentration: 150 mg P/L). The letters a, b represent the results of significance analysis of variance.

**Figure 4 ijerph-19-07227-f004:**
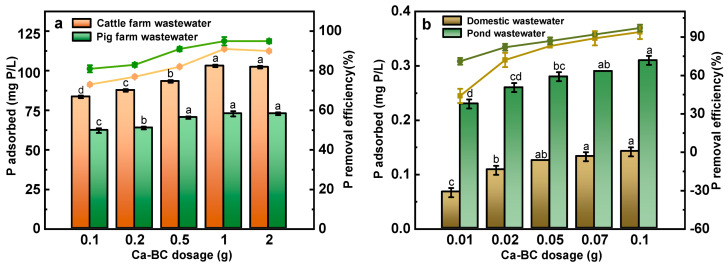
Adsorption and removal of phosphorus from cattle farm and pig farm wastewaters by Ca-BC. (**a**) Adsorption and removal of phosphorus from domestic and pond wastewaters by Ca-BC. (**b**). The letters a, b, c, d represent the results of significance analysis of variance.

**Figure 5 ijerph-19-07227-f005:**
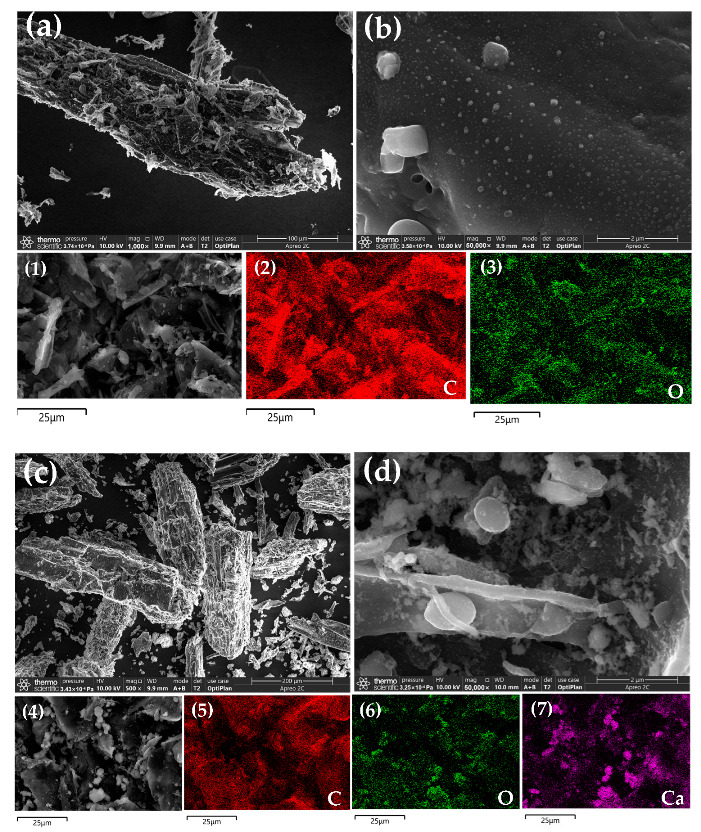
(**a**,**b**) SEM images of BC, (**c**,**d**) SEM images of Ca-BC before adsorption, (**e**,**f**) SEM images of Ca-BC after adsorption, and (1–3) EDS image of BC, (4–7) EDS image of Ca-BC before adsorption, (8–12) EDS image of Ca-BC after adsorption.

**Figure 6 ijerph-19-07227-f006:**
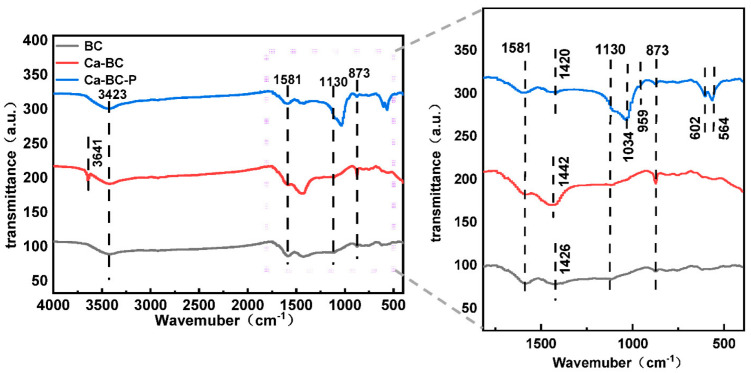
FTIR spectra of BC and Ca-BC before and after phosphorus adsorption.

**Figure 7 ijerph-19-07227-f007:**
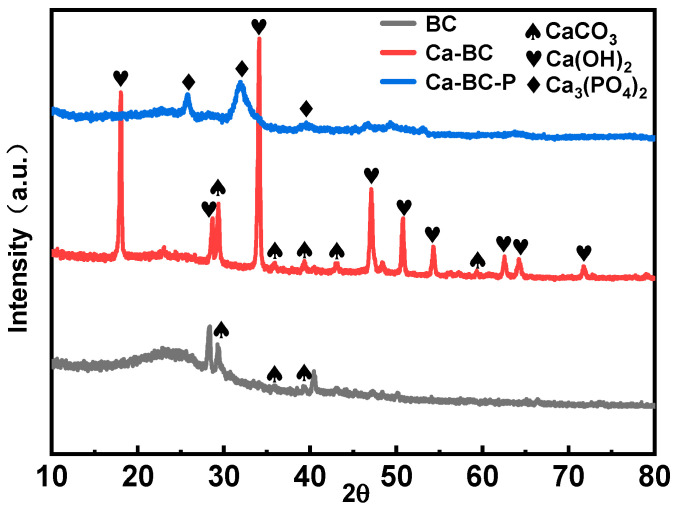
XRD spectra of BC, Ca-BC, and Ca-BC-P materials.

**Table 1 ijerph-19-07227-t001:** Fitting parameters of the phosphorus adsorption isotherms and adsorption kinetics of Ca-BC.

Langmuir			Freundlich		
*q_max_* (mg P/g)	*K_L_*	R^2^	*K_F_*	1/*n*	R^2^
88.64	0.43	0.96	29.6	0.25	0.78
**Pseudo-first order dynamic equation**			**Pseudo-second order dynamic equation**		
*q_e_* (mg P/g)	*K* _1_	R^2^	*q_e_* (mg P/g)	*K* _2_	R^2^
81.38	0.02	0.92	88.03	4.30 × 10^−4^	0.99

**Table 2 ijerph-19-07227-t002:** EDS spectrum analysis results of Ca-BC and Ca-BC-P.

	Ca-BC			Ca-BC-P			
	C	O	Ca	C	O	Ca	P
Mass percentage %	66.83 ± 2.7	21.54 ± 3.1	11.63 ± 1.3	73.30 ± 3.3	17.02 ± 2.9	2.85 ± 0.8	6.83 ± 0.9
Atomic percentage %	77.27 ± 2.4	18.70 ± 2.8	4.03 ± 1.2	82.15 ± 2.7	14.32 ± 2.3	1.24 ± 0.9	2.29 ± 0.2

**Table 3 ijerph-19-07227-t003:** Basic parameters of Ca-BC before and after preparation.

	Ca Content (mg/g)	Ca Load Efficiency (%)	P Content (mg/g)	P Load Efficiency (%)
Ca-BC	184.71 ± 5.22	18.47 ± 0.522	/	/
Ca-BC-P	146.86 ± 4.32	14.69 ± 0.432	71.60 ± 2.34	7.16 ± 0.234

## Data Availability

Not applicable.

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
