# Peer review of "Oyster Shell Modified Tobacco Straw Biochar: Efficient Phosphate Adsorption at Wide Range of pH Values"

_ijerph, 2022, doi:10.3390/ijerph19127227_

Round 1

Reviewer 1 Report

The paper is focused on the development of Ca(OH)2 loaded tobacco straw biochar for phosphate adsorption at variable pH. The topic falls within the scope of the journal. The presentation and discussion of the results could be partly improved. I recommend the publication after the following revisions:

-          An additional table with the parameters (and the corresponding errors) obtained by the fitting of the adsorption isotherms and adsorption kinetic data should be presented and properly discussed.

-          Fig. 6. The FTIR spectra look arbitrarly shifted along the y-axis. Therefore, the unit for the transmittance should be indicated as a.u. (arbitrary unit).

-          Fig. 2 reports zeta potential data at variable pH. Nevertless, experimental details for the determination of zeta potential are missed in the paragraph on the characterization.

-          The presentation and discussion of the zeta potential data should be improved.

-          Fig. 5. The scale length within SEM images is not clear. Please check and revise.

-          Introduction could be updated by evidencing that the loading of Ca(OH)2 within ecocompatible materials was explored also as antiacid for protection of lignocellulosic artwork. Related articles (Langmuir 2020, 36(14), pp. 3677-3689; ACS Applied Materials and Interfaces 2018, 10(32), pp. 27355-27364)
(Molecules. 2020, 25(19): 4502; Journal of Colloid and Interface Science, 2016, 473:1-8) could be quoted to support this consideration.  

Author Response

Reply to see the PDF.

Reviewer 2 Report

1- Its better to add novelty statement in the abstract section.

2- What are the sources of phosphrous discharge and its pollution.

3- Authors are suggested to add some latest relevant information in the introduction section.

4- Please clearly mention your research objectives.

5- Please add suitable reference for biochar production. 

6- Authors are suggested to add recent references in all of your studied parameters.

7-Its better to mention peaks value in the FTIR figure. 

8- Mention the mineral peaks in the XRD figure. 

9- Please add some useful information in the conclusion section. 

Author Response

Reply to see the PDF.

Reviewer 3 Report

Tobacco stalk and oyster shell can be a promising and cost-effective source for production of adsorbent materials. Improvement of phosphate adsorption capacity of biochar at wide pH range, such as Ca loading, has high importance. Authors used high-temperature calcination, ultrasonic treatment and stirring impregnation method for biochar fabrication. Therefore, the topic of the manuscript can be considered as interesting and relevant. The manuscript is generally well structured. Applied methods are adequate for the sample characteristics and the specific aims of the research. Materials and methods are described clearly. The manuscript contains interesting and significant results that are valuable not just for the science but also for the practice. Experimental results are discussed with relevant references. Adsorption capacity of prepared adsorber is tested by actual (real) wastewater, as well.

Comments, suggestions

Please check the typos in the manuscript (’ Ca(OH)2’ in line 13, ’ 10 and 11(concentration..’ in line 124, etc).

It is not clear how was selected/optimized the process parameters (temperature, time etc) for biochar preparation and modification (section 2.2).

Please improve the visibility of  Figure 2, 4,and 6 8mainly the axis titles).

Please provide the standard deviation for data presented in table 1 and 2.

Author Response

Reply to see the PDF.

Reviewer 4 Report

The idea of the research proposed by Menghan Feng and her team is interesting. I suggest the publication after some minor changes.

Comments:

-Abstract: the methods used for the characterization of the material can be added

-The authors should try to bring out what is the originality of the work done and why the results may have a particular interest for the scientific community

-L153-156: Please revise the information: ‘’……and surface elemental distribution of the adsorbent was observed by energy dispersive spectrometry (EDS, Zeiss Sigma 500, Germany) to observe the elemental distribution on the adsorbent surface.

-The wavelength used to measure the P concentration should be mentioned

-Please, add a comparison with other published articles

-Please verify the legend of the Table 1. What ‘’ LN-WB’’ does means?

-Please, verify Reference 3

-Please, revise the typo:

-L4-L5: Supscript the comma after the name of each author

-L13, L16: Subscript Ca(OH)2

-L34, L172: Please, add a point at the end of the sentence

-L104: Please, replace the point after ‘’below’’ with ‘’:’’

-Please, decrease the font of the Eq. 1 and Eq. 2

-Please, revise all the manuscript body. There are some spaces that must be removed or added. For example:

   -L33: Remove the space between ‘’in’’ and ‘’economic

              -L41: Remove the space between the words

              -L44: Add a space between ‘’[13-16]’’ and ‘’Therefore’’

   -L60: Remove the space between ‘’acidic condition,’’ and ‘’so’’

   -L83: Remove the space between ‘’2 h.’’ and ‘’ On the other hand’’

  -L96: Remove the space between ‘’individually.’’ and ‘’Then’’

Author Response

Reply to see the PDF.

Round 2

Reviewer 1 Report

The paper was correctly revised according to the reviewers' suggestions. On this basis, I recommend the publication in the present form.